# Determining an optimal case definition using mid-upper arm circumference with or without weight for age to identify childhood wasting in the Philippines

**Lyle Daryll Dimaano Casas**[1,2]*, **Jhanna Uy**[1,3], **Eldridge Ferrer**[4], **Charmaine Duante**[4], **Paluku Bahwere**[6], **Rene Gerard Galera, Jr.**[5], **Alice Nkoroi**[5], **Behzad Noubary**[5], **Mueni Mutunga**[6], **Sanele Nkomani**[6], **Roland Kupka**[6], **Valerie Gilbert Ulep**[1,7]

1 Research Department, Philippine Institute for Development Studies, Quezon City, Metro Manila, Philippines, 2 College of Public Health, University of the Philippines Manila, Manila, Metro Manila, Philippines, 3 Health Sciences Program, School of Science and Engineering, Ateneo de Manila University, Quezon City, Metro Manila, Philippines, 4 Nutritional Assessment and Monitoring Division, Food and Nutrition Research Institute, DOST Compound, Taguig, Metro Manila, Philippines, 5 Philippine Country Office, United Nations Children's Fund, Mandaluyong City, Metro Manila, Philippines, 6 East Asia and Pacific Regional Office, United Nations Children's Fund, Pra Nakhon, Bangkok, Thailand, 7 Ateneo Graduate School of Business, Ateneo de Manila University, Makati City, Philippines

* lcasas@pids.gov.ph

**Data Availability Statement:** De-identified data cannot be publicly shared due to legal restrictions under the Philippine government's data-sharing

## Abstract

In resource-limited areas, where accurate weight-for-height Z-scores are hard to obtain, Mid-Upper Arm Circumference (MUAC) is a simple tool to identify wasted children. MUAC alone, however, may miss identification of many wasted children, leading to untimely intervention and potentially death. Our study aimed to identify the best-performing case definition to detect wasting by Weight-for-Height z-scores (WHZ) in Filipino children aged 6–59 months. We analyzed the 2018–2019 Expanded National Nutrition Survey to assess the diagnostic performance of MUAC cutoffs and a case definition combining MUAC and weight-for-age z-score (WAZ) in identifying moderate and severe wasting compared to the WHZ criterion. The optimal cutoff and case definition was identified as having the highest area under the receiver operating characteristic curve (AUROC). Our findings showed that the current MUAC cutoffs poorly identify severe (sensitivity: 13%; specificity: 99%; AUROC: 0.558) and moderate (sensitivity: 22%; specificity: 96%; AUROC: 0.586) wasting (N = 30,522) in Filipino children. Instead, the optimal MUAC cutoff for severe and moderate wasting were <13.6cm (sensitivity: 62%; specificity: 76%; AUROC: 0.690) and 14.0cm (sensitivity: 80%; specificity: 67%; AUROC: 0.737). There was no effect of sex on MUAC cutoffs, but cutoffs increased with age. We found that the combination of WAZ < -2 or MUAC ≤ 11.7cm (Sensitivity: 80%; Specificity: 80%; AUROC: 0.800) for severe wasting and WAZ < -2 or MUAC ≤ 12.7cm (Sensitivity: 84%; Specificity: 78%; AUROC: 0.810) for moderate wasting significantly improved sensitivity for acceptable decreases in specificity. In summary, implementing alternative case definitions solely based on expanding MUAC insufficiently improves diagnostic accuracy for identifying wasted children by WHZ criteria. Combining WAZ with MUAC could increase the number of eligible children identified and treated by the

policies. These data include anthropometric, sociodemographic, and dietary survey components of the Philippine Expanded National Nutrition Survey 2018 and 2019. The datasets are owned by a third-party organization, and access through a remote server is subject to a legally binding data-sharing agreement with the Philippine Food and Nutrition Research Institute (FNRI). Access to the datasets can be requested through the FNRI Director (dostfnri47@fnri.dost.gov.ph).

**Funding:** This study was supported by the UNICEF Philippines through the Philippine Institute for Development Studies. Staff members and consultants of UNICEF were involved in the conceptualization of the study, review, and editing of the manuscript draft.

**Competing interests:** The authors have declared that no competing interests exist.

Philippine Integrated Management of Acute Malnutrition. Further studies are advised to understand the practicality and cost-effectiveness of using the proposed alternative case definitions in the Philippines.

## Introduction

Globally, wasting or acute malnutrition affects 45 million children under-five years of age [1, 2]. It kills one million children under-five annually [3]. Early identification and treatment of wasting is a core survival intervention for children that aims to prevent premature death and lifelong disability [4–7]. Early identification of wasting is particularly crucial during the first 1000 days of a child's life, the period covering pregnancy until the first two years of age, since malnutrition negatively affects child development and causes physical and mental impairments that are carried into adulthood [4–7].

In identifying wasting in children 6–59 months, the World Health Organization (WHO) recommends using either criteria: (a) the weight-for-height z-score (WHZ) and (b) Mid-Upper Arm Circumference (MUAC) [2, 4]. The first criterion, however, is difficult to use in resource-limited areas as trained health workers are in short supply and anthropometric equipment may be difficult to bring to communities and households or are only present in health facilities not easily accessible to children and mothers. Consequently, when accurate weight-for-height Z-scores are difficult to obtain, the Mid-Upper Arm Circumference (MUAC) is an alternative simple tool used to identify wasted children in need of life-saving treatment [8, 9]. The MUAC is measured using a color-coded tape that can be intuitively used by community health workers, and caregivers to assess if a child is wasted or underweight for their length or height [9].

Researchers, however, argue that relying on MUAC alone or independently may overlook a significant number of wasted children at high risk of near-term death who would miss timely intervention [10–18]. The current MUAC cutoffs recommended by the WHO exhibit varying sensitivities and specificities in different settings, with limited performance in identifying severe (6–20% sensitivity, 91–99% specificity) and moderate wasting (13–23% sensitivity, 98–99% specificity) at cutoffs of 11.5cm (severe) and 12.5cm (moderate). With a high false negative rate, these globally recommended MUAC cutoffs perform poorly in identifying children who are truly wasted based on weight-for-height z-score [12–15, 17, 18]. To improve the performance of the MUAC, researchers advocate slightly liberalizing the cutoffs and making them more suitable for specific populations. Studies have also proposed optimal MUAC cutoffs ranging from 12.5–13.4cm for severe wasting, and 13.2–13.8cm for moderate wasting, with adjustments for age and sex [12–15, 17, 18]. Additionally, a systematic review done by Khara et al. (2023) also revealed that combining case definitions with MUAC, specifically weight-for-age may be better at predicting mortality in children [19].

In the Philippines, there is great interest in using the MUAC as an innovative and affordable tool to expand access to and coverage of the Philippine Integrated Management for Acute Malnutrition Program (PIMAM) [20], the national protocol for preventing, detecting, and managing moderate and severe acute wasting [4]. Measurement using MUAC can be performed by caregivers, promoting early detection of wasting in their children [9]. However, as child body morphologies may vary across geographic regions, globally determined cutoffs may not be exactly applicable to Filipino children and in the Philippines where there is high co-existence of stunting and wasting. Consequently, improving the utility of the MUAC as a screening tool may aid the PIMAM in meeting the SDG target of 3.7% prevalence of wasting

for children under five by 2030, for which progress has plateaued for the past 30 years (World Health Organization 2014).

To this end, this study aimed to identify the best performing case definition (MUAC with or without WAZ) to accurately detect wasting in Filipino children aged 6–59 months, accounting for child characteristics that include age, sex, weight, presence of stunting, wealth quintile, and rural residence.

## Materials and methods

We used secondary data from the 2018 and 2019 Expanded National Nutrition Survey (ENNS), a national cross-sectional survey implemented by the Philippine Department of Science and Technology–Food and Nutrition Research Institute (DOST-FNRI). The ENNS is the only national-scale survey in the Philippines of its kind, and the 2018 and 2019 rounds are the latest data available. It employed a two-stage cluster sampling design using the 2013 Master Sample of the Philippine Statistics Authority, with *barangays* or villages as primary sampling units followed by the selection of secondary sampling units composed of households [21]. The ENNS collected data from 80 of the 117 provinces and highly urbanized cities in the Philippines for a nationally representative sample for each round for a total of 325,512 individuals from 94,999 households (52.4% rural and 47.6% urban) [21]. The survey covers the following components: socio-economic; anthropometric; biochemical; clinical and health; dietary; food security; maternal health; nutrition, infant and young child feeding (IYCF) practices; and government program participation.

For anthropometric measurements, the ENNS deployed trained nutritionist-dietitians and other allied health professionals to collect the height, weight, and MUAC of study participants following standard operating protocols [21]. Standing height, measured with a stadiometer (Seca GmbH & Co. KG), was collected for children aged 2 years and above, while recumbent length was measured using a medical plastic infantometer for those under 2 years. Weight measurements utilized a double-window digital scale, and for children requiring assistance, the guardian carried the child before measuring their weight. MUAC was measured using two-meter non-stretchable tapes (Seca GmbH & Co. KG). All measurements were taken twice to the nearest 0.1 centimeter, with a third reading conducted if the difference between the first two measurements exceeded 0.5cm.

For the data analysis, a total of 3,570 children with incomplete anthropometric information were excluded to come up with a final sample of 30,522 children, aged 6–59 months. To assess the accuracy of various MUAC cutoffs, we used weight-for-height z-scores (WHZ) as the reference standard. Children are classified as moderately or severely wasted given the cutoffs for WHZ and MUAC according to the WHO Child Growth Standards. We classified children as severely wasted when a child's WHZ z-score is less than –3 standard deviations (SD) below the median and moderately wasted when the WHZ z-score is less than –2SD but greater than or equal to -3SD. Using MUAC, a child is severely wasted if his or her MUAC measures less than or equal to 11.5cm and moderately wasted if his or her MUAC is above 11.5cm but less than or equal to 12.5cm.

We calculated the sensitivity, specificity, positive and negative predictive values, and area under the receiver-operator characteristic (AUROC) curve of MUAC cutoffs from 11.0cm to 14.9cm in increments of 0.1cm. Sensitivity (Specificity) was calculated by dividing the number of individuals with a positive (negative) test result using MUAC by the number of individuals who are identified as wasted (not wasted) using WHZ. Positive (negative) predictive values, on the other hand, are the proportion of positive (negative) cases that are truly positive (negative) [22]. Moreover, we assessed MUAC's performance within child subpopulations: age categories (4–5, 6–23, 24–59 months), sex, presence of stunting (calculated using height-for-age

-2SD < Z ≤ -3SD (for moderate stunting) and Z < -3SD (for severe stunting), underweight status (weight-for-age), wealth quintile, and rural or urban residence. Lastly, we also evaluated MUAC's performance when used with underweight status; that is, children are classified as moderately wasted if weight-for-age z-scores are -2SD < Z ≤ -3SD and severely wasted if Z < -3SD. Furthermore, we also assessed the characteristics (age, sex, underweight status, stunting status, dietary intake, diversity score, minimum meal frequency, and minimum acceptable diet) of children incorrectly classified as wasted vis-à-vis WHZ to provide additional information in the discussion (See **S2 File**). Full details on the data collection methodology employed for the dietary component can be found in the survey report [21].

Proposed optimal cutoffs were selected based on the highest AUROC. As the AUROC measures the overall performance of a diagnostic test, it can be used as a criterion to measure the discriminative ability of a test. AUROC represents the average specific value across all possible values, an AUROC of 1 means perfect accuracy, while an AUROC of 0.5 would mean that the test has little to no discriminative ability. Therefore, a higher AUROC value closer to 1 reflects superior overall test performance [23]. Cutoffs with AUROC values of 0.5–0.7 were interpreted as having poor diagnostic performance, 0.7–0.8 as acceptable, 0.8–0.9 as excellent, and 0.9–1.0 as outstanding [24, 25].

All analyses were conducted using STATA MP 16.0.

The authors accessed anonymized public use files of the Expanded National Nutrition Survey (ENNS) on a secured remote server through a data-sharing agreement with the FNRI. Before the implementation of the ENNS, its protocol was submitted to the Food and Nutrition Research Institute Institutional Ethics Review Committee (FNRIEC) which subsequently approved the procedures and implementation of the ENNS following the guidelines in the Declaration of Helsinki with approval code FIERC-2017-017 [21]. Written informed consent of participants was gathered by the ENNS researchers before being enrolled in the survey. The respondents' participation is voluntary, and they can refuse to participate.

## Results

### Sample characteristics and wasting prevalence

In this analysis, 30,522 children aged 6–59 months were included (**Table 1**). The majority (70.8%) of the children fall in the older age group of 24–59 months with slightly more males (51.8%) than females. Almost one-third (32.2%) of the children were stunted and one-fourth (20.4%) were underweight for their age. Over half (57.9%) of the children were from the poorest 40% income households, and most (65.9%) lived in rural areas. Overall, the MUAC identified slightly more children as moderately and severely wasted compared with the WHZ: 5.3% and 1.2% of the children were moderately and severely wasted using current MUAC cutoffs, 4.6%, and 1.0% using WHZ, and 9.0% and 2.1% using either criterion. Looking at age groups, the current MUAC cutoffs identified more wasted children in the 6–23 months age group, while the WHZ identified more in the older age group of 24–59 months. Across sex, MUAC identified more wasted children among females whereas WHZ identified more among males. In terms of stunting and underweight status, the MUAC identified more severely wasted children among those who are also stunted, while the WHZ identified more moderately and severely wasted children among those who are also underweight.

### Performance of current global MUAC cutoffs in identifying moderate and severe wasting

Overall, the current MUAC cutoffs exhibited poor performance in identifying Filipino children aged 6–59 months with moderate (AUROC: 0.586) and severe wasting (AUROC: 0.557).

**Table 1. Characteristics and prevalence of moderate and severe wasting among children 6–59 months in the ENNS, 2018–2019.**

| Variable | n, (%) | Moderate Wasting (n, %) | | | | | | Severe Wasting (n, %) | | | | | |
|---|---|---|---|---|---|---|---|---|---|---|---|---|---|
| | | WHZ (-2SD < Z ≤ -3SD) | | MUAC (>11.5 and ≤ 12.5) | | WHZ (-2SD < Z ≤ -3SD) or MUAC (>11.5 and ≤ 12.5) | | WHZ (Z < -3SD) | | MUAC (≤ 11.5) | | WHZ (Z < -3SD) or MUAC (≤ 11.5) | |
| | | n (%) | Prevalence within sub-group | n (%) | Prevalence within sub-group | n (%) | Prevalence within sub-group | n (%) | Prevalence within sub-group | n (%) | Prevalence within sub-group | n (%) | Prevalence within sub-group |
| **All children** | **30,522** | **1419** | **4.6%** | **1626** | **5.3%** | **2737** | **9.0%** | **309** | **1.0%** | **366** | **1.2%** | **636.00** | **2.1%** |
| *Age* | | | | | | | | | | | | | |
| 6–23 months | 8924 (29.2) | 518 (36.5) | 1.7% | 1149 (70.7) | 3.8% | 1475 (53.9) | 4.8% | 135 (43.7) | 0.4% | 268 (73.2) | 0.9% | 373 (58.6) | 1.2% |
| 24–59 months | 21598 (70.8) | 901 (63.5) | 3.0% | 477 (29.3) | 1.6% | 1262 (46.1) | 4.1% | 174 (56.3) | 0.6% | 98 (26.8) | 0.3% | 263 (41.4) | 0.9% |
| *Sex* | | | | | | | | | | | | | |
| Male | 15818 (51.8) | 815 (57.4) | 2.7% | 652 (40.1) | 2.1% | 1311 (47.9) | 4.3% | 184 (59.5) | 0.6% | 155 (42.3) | 0.5% | 313 (49.2) | 1.0% |
| Female | 14704 (48.2) | 604 (42.6) | 2.0% | 974 (59.9) | 3.2% | 1426 (52.1) | 4.7% | 125 (40.5) | 0.4% | 211 (57.7) | 0.7% | 323 (50.8) | 1.1% |
| *Height-for-age z-score (stunting status)* | | | | | | | | | | | | | |
| 2SD < z ≤ -2SD | 20680 (67.8) | 781 (55) | 2.6% | 782 (48.1) | 2.6% | 1423 (52) | 4.7% | 218 (70.6) | 0.7% | 157 (42.9) | 0.5% | 359 (56.4) | 1.2% |
| -2SD < z ≤ -3SD | 2721 (8.9) | 222 (15.6) | 0.7% | 330 (20.3) | 1.1% | 475 (17.4) | 1.6% | 47 (15.2) | 0.2% | 102 (27.9) | 0.3% | 132 (20.8) | 0.4% |
| ≤ -3SD | 7121 (23.3) | 416 (29.3) | 1.4% | 514 (31.6) | 1.7% | 839 (30.7) | 2.7% | 44 (14.2) | 0.1% | 107 (29.2) | 0.4% | 145 (22.8) | 0.5% |
| *Weight-for-age z-score (underweight status)* | | | | | | | | | | | | | |
| 2SD < z ≤ -2SD | 23841 (78.1) | 289 (20.4) | 0.9% | 757 (46.6) | 2.5% | 1007 (36.8) | 3.3% | 61 (19.7) | 0.2% | 143 (39.1) | 0.5% | 203 (31.9) | 0.7% |
| -2SD < z ≤ -3SD | 1070 (3.5) | 400 (28.2) | 1.3% | 260 (16) | 0.9% | 526 (19.2) | 1.7% | 148 (47.9) | 0.5% | 110 (30.1) | 0.4% | 227 (35.7) | 0.7% |
| ≤ -3SD | 5169 (16.9) | 730 (51.4) | 2.4% | 606 (37.3) | 2.0% | 1201 (43.9) | 3.9% | 100 (32.4) | 0.3% | 112 (30.6) | 0.4% | 205 (32.2) | 0.7% |
| *Wealth quintile* | | | | | | | | | | | | | |
| Poorest | 10227 (33.6) | 600 (42.5) | 2.0% | 760 (46.7) | 2.5% | 1205 (44.1) | 4.0% | 145 (46.9) | 0.5% | 169 (46.2) | 0.6% | 294 (46.2) | 1.0% |
| Poor | 7407 (24.3) | 370 (26.2) | 1.2% | 407 (25) | 1.3% | 701 (25.7) | 2.3% | 75 (24.3) | 0.2% | 105 (28.7) | 0.3% | 171 (26.9) | 0.6% |
| Middle | 5516 (18.1) | 219 (15.5) | 0.7% | 245 (15.1) | 0.8% | 428 (15.7) | 1.4% | 39 (12.6) | 0.1% | 44 (12) | 0.1% | 77 (12.1) | 0.3% |
| Richer | 4236 (13.9) | 132 (9.3) | 0.4% | 141 (8.7) | 0.5% | 247 (9) | 0.8% | 33 (10.7) | 0.1% | 30 (8.2) | 0.1% | 59 (9.3) | 0.2% |
| Richest | 3082 (10.1) | 92 (6.5) | 0.3% | 73 (4.5) | 0.2% | 150 (5.5) | 0.5% | 17 (5.5) | 0.1% | 18 (4.9) | 0.1% | 35 (5.5) | 0.1% |
| *Type of Residence* | | | | | | | | | | | | | |
| Rural | 20108 (65.9) | 992 (69.9) | 3.3% | 1146 (70.5) | 3.8% | 1921 (70.2) | 6.3% | 202 (65.4) | 0.7% | 251 (68.6) | 0.8% | 427 (67.1) | 1.4% |
| Urban | 10414 (34.1) | 427 (30.1) | 1.4% | 480 (29.5) | 1.6% | 816 (29.8) | 2.7% | 107 (34.6) | 0.4% | 115 (31.4) | 0.4% | 209 (32.9) | 0.7% |

Source: Authors' analysis of pooled 2018–2019 Expanded National Nutrition Survey data (Department of Science and Technology—Food and Nutrition Research Institute).

Note: ENNS = Expanded National Nutrition Survey; WHZ = Weight-for-height z-score; MUAC = Mid-Upper Arm Circumference

**Table 2. Diagnostic performance of current MUAC cutoffs in the identification of moderate wasting diagnosed using WHZ Z-scores among children aged 6 to 59 months.**

| MUAC (≥ 11.5 to <12.5cm) | Weight for Height/Length Z-score (-2SD < Z ≤ -3SD) | | Total |
|---|---|---|---|
| | Positive | Negative | |
| Positive | 308 | 1,318 | 1,626 |
| Negative | 1,111 | 27,785 | 28,896 |
| Total | 1,419 | 29,103 | 30,522 |

Note: Sensitivity: 21.71%, Specificity: 95.47%, AUROC: 0.586

Specifically, the current cutoffs have high specificity, but very low sensitivity. The cutoffs were 22% sensitive and 96% specific for moderate wasting (**Table 2**) and 13% sensitive and 99% specific for severe wasting (**Table 3**). This indicates that only 22% and 13% of moderately and severely wasted children were correctly identified as wasted by the current MUAC cutoffs. On the other hand, 96% and 99% of children who were truly not moderately and severely wasted were correctly ruled out.

## Performance of proposed optimal MUAC cutoffs and case definitions combining MUAC and WAZ for moderate and severe wasting

**Tables 4** and **5** show the AUROC, sensitivity, specificity, PPV, and NPV of the current and proposed optimal MUAC cutoffs. The detailed information (full tables) of the determination of the optimal cutoffs can be found in **Tables 1** and **2 in S1 File.**

In general, identified optimal MUAC cutoffs calculated for all children and various subgroups were higher and with acceptable performance (based on AUROC) in this sample of children than the current global cutoffs recommended by the WHO (Tables 4 and 5, Figs 1 and 2). For moderate wasting, the optimal MUAC cutoff was 14.0cm (AUROC 0.741), with a sensitivity of 80% and specificity of 68%, with a PPV of 11.0% and NPV of 98.6% (**Table 4**, Fig 1, and Table 1 in **S1** File). This means that the MUAC at < 14.0cm correctly identified 80% of children who were moderately wasted and 68% as not moderately wasted by weight-for-height criteria. For severe wasting, the optimal MUAC cutoff was <13.6cm (AUROC 0.690), demonstrating a sensitivity of 62% and specificity of 76%, with a PPV of 2.6% and NPV of 99.5%. **(Table 5, Fig 2, and Table 2 in S1 File**).

In practical terms, MUAC at <13.6cm correctly identified 62% of children who were severely wasted and 76% as not severely wasted. Both optimal MUAC cutoffs for moderate and severe wasting exhibited higher negative predictive values at >98% and low PPV at 11.0% and 2.6%.

**Table 3. Diagnostic performance of current MUAC cutoffs in the identification of severe wasting diagnosed using WHZ Z-scores among children aged 6 to 59 months.**

| MUAC (≤ 11.5) | Weight for Height/Length Z-score (Z < -3SD) | | Total |
|---|---|---|---|
| | Positive | Negative | |
| Positive | 39 | 327 | 366 |
| Negative | 270 | 29,886 | 30,156 |
| Total | 309 | 30,213 | 30,522 |

Note: Sensitivity: 12.62%, Specificity: 98.92%, AUROC: 0.557

**Table 4. Diagnostic performance of current and identified optimal MUAC cutoffs and evaluated alternative case definitions for moderate wasting.**

| Categories | Moderate Wasting | | | | | |
|---|---|---|---|---|---|---|
| | Cutoff (cm) | AUROC (LB-UB) | Sensitivity | Specificity | PPV | NPV |
| *All children* | Current ($\geq$ 11.5 to <12.5cm) | 0.586 (0.575–0.597) | 21.71% | 95.47% | 18.94% | 96.16% |
| | <14.0 | 0.741 (0.730–0.751) | 79.63% | 68.47% | 10.97% | 98.57% |
| | <13.9 or WAZ <-3 | 0.756 (0.746–0.767) | 81.61% | 69.63% | 11.58% | 98.73% |
| | <12.4 or WAZ <-2 | 0.815 (0.805–0.825) | 83.23% | 79.79% | 16.72% | 98.99% |
| *Age (months)* | | | | | | |
| 6–23 m | ($\geq$ 11.5 to <12.5cm) | 0.628 (0.607–0.649) | 37.07% | 88.62% | 16.71% | 95.81% |
| | <13.2 | 0.741 (0.723–0.759) | 80.12% | 68.03% | 13.38% | 98.23% |
| 24–59 | ($\geq$ 11.5 to <12.5cm) | 0.556 (0.545–0.567) | 12.87% | 98.26% | 24.32% | 96.28% |
| | <14.1 | 0.751 (0.737–0.766) | 75.25% | 75.03% | 11.60% | 98.58% |
| *Sex* | | | | | | |
| Male | ($\geq$ 11.5 to <12.5cm) | 0.579 (0.566–0.593) | 19.14% | 96.69% | 23.93% | 95.65% |
| | <14.0 | 0.748 (0.734–0.763) | 78.40% | 71.25% | 12.90% | 98.38% |
| Female | ($\geq$ 11.5 to <12.5cm) | 0.597 (0.579–0.614) | 25.17% | 94.17% | 15.61% | 96.71% |
| | <13.9 | 0.737 (0.720–0.753) | 80.13% | 67.23% | 9.48% | 98.75% |
| *Weight-for-age z-score (underweight status)* | | | | | | |
| $\leq$ -3SD (Severe) | ($\geq$ 11.5 to <12.5cm) | 0.573 (0.546–0.601) | 33.50% | 81.19% | 51.54% | 67.16% |
| | <12.7 | 0.589 (0.558–0.619) | 52.25% | 65.52% | 47.50% | 69.68% |
| -2SD < z $\leq$ -3SD (Moderate) | ($\geq$ 11.5 to <12.5cm) | 0.539 (0.525–0.554) | 18.49% | 89.39% | 22.28% | 86.96% |
| | <13.7 | 0.624 (0.607–0.642) | 73.56% | 51.27% | 19.89% | 92.18% |
| 2SD < z $\leq$ -2SD (Normal | ($\geq$ 11.5 to <12.5cm) | 0.552 (0.532–0.572) | 13.49% | 96.95% | 5.15% | 98.92% |
| | <14.0 | 0.693 (0.665–0.720) | 64.01% | 74.50% | 2.99% | 99.41% |
| *Height-for-age z-score (stunting status)* | | | | | | |
| $\leq$ -3SD (Severe) | ($\geq$ 11.5 to <12.5cm) | 0.623 (0.591–0.655) | 34.68% | 89.88% | 23.33% | 93.94% |
| | <13.1 | 0.715 (0.684–0.747) | 69.37% | 73.67% | 18.97% | 96.44% |
| -2SD < z $\leq$ -3SD (Moderate) | ($\geq$ 11.5 to <12.5cm) | 0.578 (0.558–0.598) | 21.88% | 93.69% | 17.70% | 95.08% |
| | <13.8 | 0.726 (0.706–0.746) | 79.57% | 65.68% | 12.58% | 98.11% |
| 2SD < z $\leq$ -2SD (Normal) | ($\geq$ 11.5 to <12.5cm) | 0.573 (0.560–0.587) | 17.93% | 96.77% | 17.90% | 96.78% |
| | <14.0 | 0.752 (0.737–0.767) | 75.93% | 74.49% | 10.46% | 98.75% |

Note: AUROC = Area Under the Receiver-Operator Characteristic Curve; LB = Lower bound; UB = Upper bound; PPV = Positive predictive value; NPV = Negative predictive value

The same general trend of higher MUAC cutoffs (13.0–14.5cm), better performance (0.650–0.751, acceptable), but low PPV (0.6–46.8%) and high NPV (69.7–99.8%) compared to current cutoffs were found in calculations for child subpopulations. The optimal cutoffs for moderate and severe wasting for the age groups of 6–23, and 24–59 months are 13.2cm and 13.0cm, and 14.1cm and 14.5cm, showing an increase in the cutoff as age increases. (Tables 4 and 5). Looking at sex, the optimal cutoffs for moderate and severe wasting in male children were 14.0cm and 13.7cm, and in females 13.9cm and 13.4cm.

Lastly, we found that classifying children as wasted using either MUAC or weight-for-age resulted in optimal cutoffs close to global cutoffs with high sensitivities and specificities when calculated among all children. For moderate wasting, the optimal cutoff was 12.4cm or WAZ <-2 (AUROC: 0.815, Sensitivity: 83.23%, Specificity: 79.79%, PPV: 16.72%, NPV: 98.99%). For severe wasting, the optimal cutoff was 11.6cm or WAZ <-2 (AUROC: 0.801, Sensitivity: 80.58%, Specificity: 79.62%, PPV: 3.89%, NPV: 99.75%).

**Table 5. Diagnostic performance of current and identified optimal MUAC cutoffs and evaluated alternative case definitions for severe wasting.**

| Categories | Severe Wasting | | | | | |
|---|---|---|---|---|---|---|
| | Cutoff (cm) | AUROC (LB-UB) | Sensitivity | Specificity | PPV | NPV |
| *All children* | Current (<11.5cm) | 0.558 (0.539–0.576) | 12.62% | 98.92% | 10.66% | 99.10% |
| | <13.6 | 0.690 (0.662–0.717) | 61.81% | 76.09% | 2.58% | 99.49% |
| | <13.4 or WAZ <-3 | 0.752 (0.727–0.777) | 72.17% | 78.26% | 3.28% | 99.64% |
| | <11.6 or WAZ <-2 | 0.801 (0.779–0.823) | 80.58% | 79.62% | 3.89% | 99.75% |
| *Age (months)* | | | | | | |
| 6–23 m | <11.5cm | 0.598 (0.562–0.633) | 22.22% | 97.29% | 11.19% | 98.79% |
| | <13.0 | 0.650 (0.607–0.692) | 55.56% | 74.40% | 3.23% | 99.09% |
| 24–59 | <11.5cm | 0.524 (0.507–0.540) | 5.17% | 99.58% | 9.18% | 99.23% |
| | <14.5 | 0.698 (0.667–0.729) | 78.16% | 61.47% | 1.62% | 99.71% |
| *Sex* | | | | | | |
| Male | <11.5cm | 0.567 (0.541–0.592) | 14.13% | 99.17% | 16.77% | 98.99% |
| | <13.7 | 0.704 (0.669–0.738) | 65.22% | 75.49% | 3.04% | 99.46% |
| Female | <11.5cm | 0.545 (0.518–0.572) | 10.40% | 98.64% | 6.16% | 99.23% |
| | <13.4 | 0.683 (0.641–0.726) | 63.20% | 73.48% | 2.00% | 99.57% |
| *Weight-for-age z-score (underweight status)* | | | | | | |
| ≤ -3SD (Severe) | <11.5cm | 0.562 (0.528–0.596) | 20.95% | 91.43% | 28.18% | 87.81% |
| | <12.0 | 0.588 (0.547–0.628) | 35.14% | 82.43% | 24.30% | 88.79% |
| -2SD < z ≤ -3SD (Moderate) | <11.5cm | 0.525 (0.499–0.550) | 7.00% | 97.93% | 6.25% | 98.16% |
| | <13.6 | 0.564 (0.516–0.613) | 60.00% | 52.87% | 2.45% | 98.53% |
| 2SD < z ≤ -2SD (Normal) | <11.5cm | 0.505 (0.489–0.521) | 1.64% | 99.40% | 0.70% | 99.75% |
| | <14.2 | 0.611 (0.548–0.673) | 55.74% | 66.37% | 0.42% | 99.83% |
| *Height-for-age z-score (stunting status)* | | | | | | |
| ≤ -3SD (Severe) | <11.5cm | 0.665 (0.595–0.734) | 36.17% | 96.82% | 16.67% | 98.85% |
| | <12.7 | 0.797 (0.738–0.857) | 78.72% | 80.70% | 6.69% | 99.54% |
| -2SD < z ≤ -3SD (Moderate) | <11.5cm | 0.561 (0.510–0.612) | 13.64% | 98.57% | 5.61% | 99.46% |
| | <13.4 | 0.707 (0.637–0.777) | 68.18% | 73.21% | 1.56% | 99.73% |
| 2SD < z ≤ -2SD (Normal | <11.5cm | 0.533 (0.516–0.551) | 7.34% | 99.31% | 10.19% | 99.02% |
| | <14.5 | 0.683 (0.655–0.711) | 77.52% | 59.05% | 1.98% | 99.60% |

Note: AUROC = Area Under the Receiver-Operator Characteristic Curve; LB = Lower bound; UB = Upper bound; PPV = Positive predictive value; NPV = Negative predictive value

## Discussion

In settings like the Philippines where equipment is scarce and measuring accurate weight-for-height z scores is challenging in the field, the MUAC is a highly practical and simple tool that can be used in the early identification of wasted children. To improve the MUAC's utility in the Philippine setting, we investigated alternative MUAC cutoffs that may better identify wasting by weight-for-height in children 6–59 months in the Philippines, accounting for child characteristics and in combination WAZ as a potential alternative case definition. Based on the AUROC score, we found that liberalizing MUAC cutoffs alone or identifying age- and sex-specific cutoffs did not sufficiently improve the diagnostic accuracy of MUAC. Instead, a combination of MUAC ≤ 12.4cm or WAZ < -2 for moderate wasting and MUAC ≤ 11.6cm or WAZ < -2 or for severe wasting were the optimal case definitions.

Our study confirms that MUAC, using current WHO cutoffs, is a highly specific, but poorly sensitive diagnostic test for identifying wasting diagnosed by WHZ in Filipino children 6–59 months. Consistent with the results in other countries [13, 14, 17, 18], using this cut-off in the

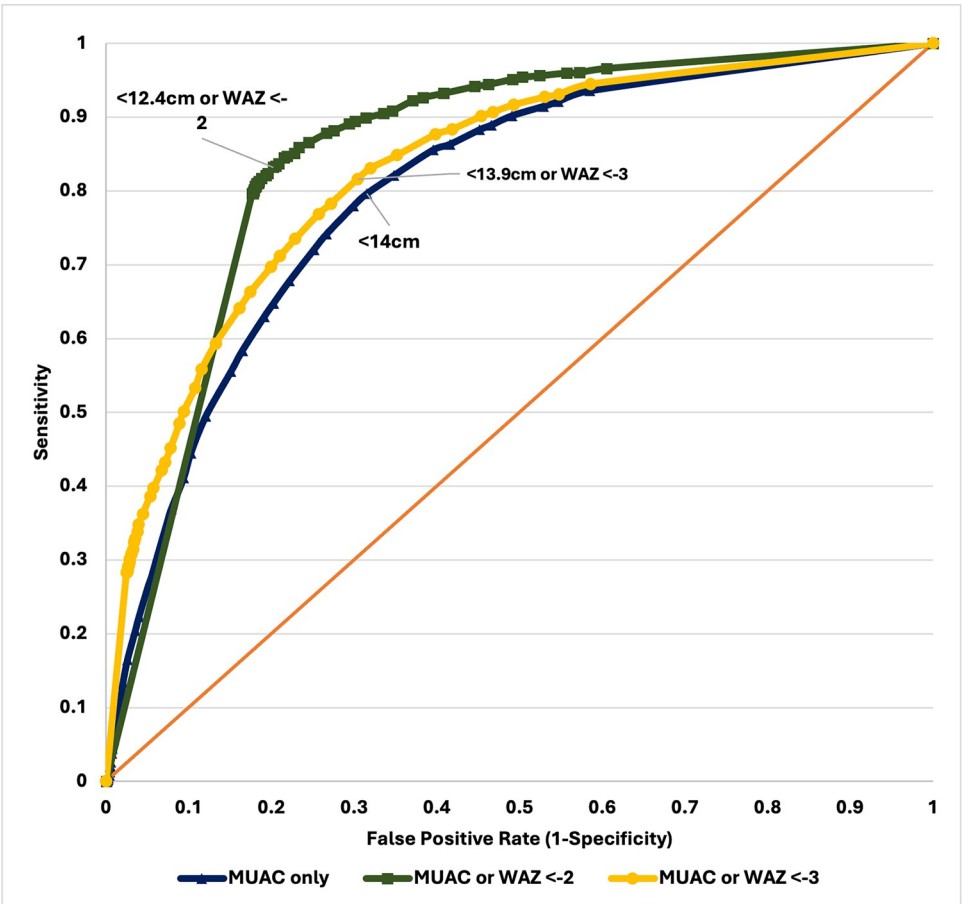

**Fig 1. ROC curve of MUAC cutoffs against weight-for height/length Z-scores (moderate wasting).**

field may lead to a high proportion of false negatives or wasted children that may be left undiagnosed and untreated. Aligning with these past studies, our findings underscore the importance of considering the specific context of the country of implementation. Generalized cutoffs may not be universally applicable due to variations in population health, genetics, or body morphology.

Less conservative MUAC cutoffs for all Filipino children and cutoffs specific to child subpopulations exhibited better but still limited diagnostic performance in terms of AUROC score (<0.8) and sensitivity in exchange for lowered specificity. This finding is consistent with findings in other countries like India, Viet Nam, Cambodia, Ethiopia, Indonesia, Pakistan, and Nepal [13, 14, 17, 26–29]. The current MUAC cutoffs showed sensitivity and specificity ranging from 6–50% and 90–99% in these settings. This is found to be similar with our findings (for severe wasting, sensitivity: 13%; specificity: 99%; AUROC: 0.558; and moderate wasting, sensitivity: 22%; specificity: 96%; AUROC: 0.586). A reason for the poor performance of the MUAC alone and difficulty in improving its accuracy is that current WHO-recommended global MUAC cutoffs poorly overlap with WHZ cutoffs. Meaning, MUAC and WHZ identify different subpopulations of children [13, 14, 17, 26–29]. In addition, having several cutoffs tailored to child demographic factors may introduce practical concerns like confusion in the field or increased workload for trained personnel that detracts them from other program duties.

Interestingly, an alternative case definition that included WAZ provided the most improvement in the AUROC and sensitivity for both moderate and severe wasting at MUAC cutoffs

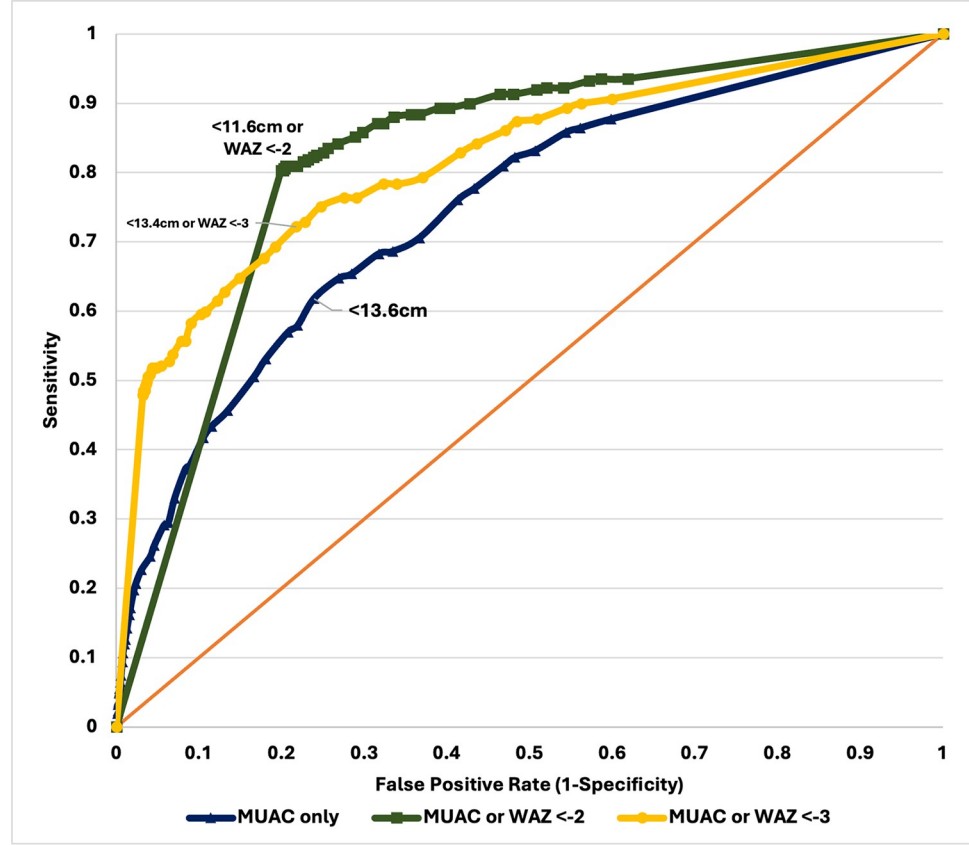

**Fig 2. ROC curve of MUAC cutoffs against weight-for height/length Z-scores (severe wasting).**

only different by 0.2cm than current global cutoffs (moderate: 12.4cm, severe: 11.6cm). Compared to the current MUAC cutoffs, the diagnostic performance of this criteria improved by 20%, and sensitivity increased from 12–20% to 80% with a specificity of around 80% as well. This offers a balance between maximizing the identification of wasted children without having too many false positives and false negatives. This is supported by findings from other studies, reporting that MUAC with WAZ was able to detect all near-term mortality associated with wasting, concurrent wasting and stunting (WaSt), and other anthropometric deficits [19, 30, 31]. This result is also consistent with studies that have shown the association of WAZ with wasting (weight-for-height), making WAZ a potential anthropometric indicator that could also predict wasting in settings where measurement of height is not possible [32, 33]. Practically, measuring weight is still accessible to caregivers, and slightly adjusting the cutoffs will not imply huge costs in the re-production of specific MUAC tapes or training of field personnel. Further, the measurement of weight and the use of weight for age is already part of the country's growth monitoring program; thus, this contributed to its integration with the PIMAM [34, 35]

One tradeoff of a much-improved sensitivity that decreases false negatives is the decrease in test specificity compared to the current cutoffs. This may result in increased false positives or admitting children who are not wasted to treatment which may potentially strain the local healthcare system. The results show that with the improvement in the identification of true positives, there is also an exponential increase in the number of false positives due to the low prevalence of wasting in the population. Nonetheless, some of these children who may be false

positive may be those at high risk of acute malnutrition or have other nutrition-related problems that need treatment. Looking at the characteristics of false positive children (S2 File), most of them are also underweight and stunted, near the WHZ threshold of being wasted, have low dietary diversity scores, and are not meeting the minimum acceptable diet. Hence, these children may benefit from the other appropriate interventions that can be identified upon contact with the health system, and which could halt the progression to more severe deterioration of nutrition status and improve the nutrition outcomes of the child. Further, the point of contact of these children to the health system may be critical–for them to receive any other health services that they may benefit from. In this context, the increase in false positives for the combined WAZ and MUAC criteria may be justified, given that the cost of a false negative or missed treatment is substantial in terms of lifelong morbidity and increased risk of mortality.

Our study has several limitations. First, the consideration of the presence of edema was not incorporated in the analyses due to limitations in the available data. The presence of bilateral pitting edema is included in the current WHO-recommended criteria for identification of acute malnutrition which may also be accounted for in the analysis if data is available. Second, children coming from indigenous populations with characteristics that might not be comparable to the general population were not accounted for in the analysis due to limitations in the data. Third, there is relatively limited data on younger infants <6 months old, thus we excluded this group in this analysis. Our results may not be externally generalizable or appropriate to apply screening among indigenous children or children <6 months old. Fourth, the results for subgroup analyses (by age, sex, underweight, and stunting status) should be interpreted carefully due to the low prevalence of severe and moderate wasting. Despite the limitations of the data, this dataset is the largest and only national survey that collects detailed anthropometric measurements for children [36]. Nevertheless, this is the first study in the Philippines that examines alternative MUAC cutoffs that may be more suitable to and accurate for Filipino children compared to simply adopting globally accepted cutoffs.

Our findings support the need to identify alternative anthropometric case definitions to the currently recommended MUAC cutoffs to maximize the identification of children wasted by weight-for-height z-score criteria in settings not using this indicator. We have provided information that contributes to the local evidence to support the efforts to refine and contextualize criteria for the identification of wasted children and promote early initiation of management of acute malnutrition and improve access and coverage for PIMAM services. An option that could be explored as suggested by a study in Cambodia is sequential testing, where children who will be identified in the field as wasted using the liberalized cutoffs in the field will further be assessed in a health facility [13]. Children who would not require admission to treatment for wasting may take that opportunity to receive other health and nutrition interventions to prevent and avoid being malnourished soon. Overall, we recommend further studies to validate findings and better understand their practical use and cost to the health system.

## Supporting information

**S1 File. Diagnostic performance of MUAC cutoffs in identifying moderate and severe wasting (S1 Table 1 and 2).**
(DOCX)

**S2 File. Characteristics of children incorrectly classified as wasted vis-à-vis weight for height z-score.**
(DOCX)

**S3 File. Sample characteristics of full representative sample and final sample.**
(DOCX)

## Author Contributions

**Conceptualization:** Lyle Daryll Dimaano Casas, Jhanna Uy, Paluku Bahwere, Rene Gerard Galera, Jr., Alice Nkoroi, Valerie Gilbert Ulep.

**Data curation:** Lyle Daryll Dimaano Casas, Eldridge Ferrer, Charmaine Duante.

**Formal analysis:** Lyle Daryll Dimaano Casas.

**Funding acquisition:** Lyle Daryll Dimaano Casas, Rene Gerard Galera, Jr., Alice Nkoroi.

**Investigation:** Lyle Daryll Dimaano Casas, Jhanna Uy, Valerie Gilbert Ulep.

**Methodology:** Lyle Daryll Dimaano Casas, Jhanna Uy.

**Project administration:** Lyle Daryll Dimaano Casas, Jhanna Uy, Valerie Gilbert Ulep.

**Resources:** Lyle Daryll Dimaano Casas, Jhanna Uy, Valerie Gilbert Ulep.

**Supervision:** Valerie Gilbert Ulep.

**Validation:** Lyle Daryll Dimaano Casas, Jhanna Uy, Paluku Bahwere, Rene Gerard Galera, Jr., Alice Nkoroi, Valerie Gilbert Ulep.

**Visualization:** Lyle Daryll Dimaano Casas.

**Writing – original draft:** Lyle Daryll Dimaano Casas, Jhanna Uy.

**Writing – review & editing:** Lyle Daryll Dimaano Casas, Jhanna Uy, Eldridge Ferrer, Charmaine Duante, Paluku Bahwere, Rene Gerard Galera, Jr., Alice Nkoroi, Behzad Noubary, Mueni Mutunga, Sanele Nkomani, Roland Kupka, Valerie Gilbert Ulep.

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
