## [Decision Letter · Decision Letter 0]

17 Oct 2024

PONE-D-24-23038Determining an optimal case definition using mid-upper arm circumference with or without weight for age to identify childhood wasting in the PhilippinesPLOS ONE

Dear Dr. Casas,

Thank you for submitting your manuscript to PLOS ONE. After careful consideration, we feel that it has merit but does not fully meet PLOS ONE’s publication criteria as it currently stands. Therefore, we invite you to submit a revised version of the manuscript that addresses the points raised during the review process.

I would like to sincerely apologise for the delay you have incurred with your submission. It has been exceptionally difficult to secure reviewers to evaluate your study. We have now received the completed reviews; the comments are available below. The reviewers have raised significant scientific concerns about the study that need to be addressed in a revision. Please pay particular attention to Reviewer#2's comments.

Please revise the manuscript to address all the reviewer's comments in a point-by-point response in order to ensure it is meeting the journal's publication criteria. Please note that the revised manuscript will need to undergo further review, we thus cannot at this point anticipate the outcome of the evaluation process.

We look forward to receiving your revised manuscript.

Kind regards,

Miquel Vall-llosera Camps

Senior Staff Editor

PLOS ONE

Reviewers' comments:

Reviewer's Responses to Questions

**Comments to the Author**

1. Is the manuscript technically sound, and do the data support the conclusions?

Reviewer #1: Yes

Reviewer #2: Partly

Reviewer #3: Yes

2. Has the statistical analysis been performed appropriately and rigorously? 

Reviewer #1: Yes

Reviewer #2: Yes

Reviewer #3: Yes

3. Have the authors made all data underlying the findings in their manuscript fully available?

Reviewer #1: Yes

Reviewer #2: Yes

Reviewer #3: No

4. Is the manuscript presented in an intelligible fashion and written in standard English?

Reviewer #1: Yes

Reviewer #2: Yes

Reviewer #3: Yes

5. Review Comments to the Author

Reviewer #1: MUAC cut offs were largely based on risk of mortality. There are several studies to suggest that MUAC less than 11.5 Cm have higher risk of mortality.

However, there is definite need for finding other criteria which has better sensitivity for screening in the community .

Reviewer #2: Dear authors,

Thank you for the opportunity to review your paper on an adjusted MUAC cut-off in combination with WAZ for identifying acute malnutrition in the Philippines. Please find my feedback on your article below:

It would be good to provide more rationale for using MUAC in combination with WAZ- WAZ not an indicator for acute malnutrition. Familiar with health workers, but potential for misclassification, especially if low WAZ is due to low HAZ. Additionally, it loses practical field advantages of WHZ and MUAC in that age in months is required for plotting/calculating WAZ but not needed for MUAC and WHZ. I disagree with the definition provided from line 138 on page 5- that a child should be classified as moderately wasted if their weight-for-age Z-score is -2SD<waz<-3, and="" if="" severely="" wasted="" waz="">Area under the ROC was used as the indicator of the best-performing MUAC cut-off value- why was something like Youden’s J not used? This would give a better indication of the balance between sensitivity and specificity- the AUROC will only give an indication of the diagnostic accuracy of the test. In table 4, raised MUAC cut-offs are recommended that raise the sensitivity of MUAC to approximately 80%, however this comes at a cost of ~30% lower specificity.

Results describe 4.6% severely wasted (WHZ) of 30522= 1419 children; 5.3% (1626 of 30522) severely wasted according to MUAC. Given the numbers presented in the study, sensitivity is 83.23% and specificity is 79.79% for the alternative MUAC, raising the cut-off would result in 1353 correctly classified as true positives, but with a corresponding increase in false positives to approximately 5800. This represents a marked improvement on the number of true positive cases identified by MUAC, but with a correspondingly exponential increase in the number of false positives identified. The much higher absolute number of false positives is due to the relatively low prevalence of the condition in the general population. This point is conceded in the discussion, and table 6 presents and interesting analysis of the misclassifications- this on its own would make for an interesting article when assessing current MUAC recommendations. Misclassification often results in eroded trust in health systems, opportunity costs for caregivers and a higher burden of health systems, which needs to be considered in this paper. While I agree that nutritionally vulnerable children would benefit from earlier intervention, treatment for severe and moderate acute malnutrition is markedly different from treatment for stunting and related underweight for age. For example, improving dietary diversity as mentioned as a risk factor in these children in the paper may be a good way to treat stunting, and possibly underweight, but will not be affected by programmes treating acute malnutrition with F-75 and F-100 based products.

Overall it is an interesting idea, however, more consideration needs to be given to the practical implications of the changed MUAC cut-off values presented.</waz<-3,>

Reviewer #3: This manuscript describes a secondary analysis of data from a national survey to identify alternative MUAC cut-offs to better identify wasting in young children in the Philippines. Overall this paper is well written and logical with outcomes that, while not perfect, may offer an opportunity for improved identification of wasting in this vulnerable group.

P4, line 111 - were potential participants able to decline to participate?

P5, line 124 - was the sample still representative after removal of incomplete cases? Was there anything about this group that may have been different? eg more remote? Does that matter?

P5, line 133 - need refs in this sentence for the method of calculating Sn, Sp, PPV and NPV.

Table 1 - add (months) to age; double check % within subgroups as some do not appear to be correct. If I am misinterpreting I suggest that the table should be reformatted to be clearer.

P9, line 175 - please define the cut-offs for performance for your AUROC. Probably best placed in the methods but if you are going to say they performed poorly what is the criterion for performance.

Tables 2 and 3 - indication of how Sn and Sp were calculated would help your reader.

P13, line 234 - I am not clear why this section or Table 6 are included. I cannot see in the methods that this was planned and there appears to be no mention of diet in the methods at all. Similarly the discussion of this content. If it is to be included there needs to be better alignment through the methods and ideally the same content provided for all children for more transparent comparison.

6. PLOS authors have the option to publish the peer review history of their article (what does this mean?). If published, this will include your full peer review and any attached files.

Reviewer #1: **Yes: **DR PRAVEEN KUMAR

Reviewer #2: No

Reviewer #3: No

---

## [Author Response · Author response to Decision Letter 0]

22 Oct 2024

We have addressed the comments on a point-by-point basis in the following pages, noting the places where revisions to the manuscripts have been made based on these comments. We thank the reviewers for their time in reading the manuscript again thoroughly and providing additional feedback to improve the quality of our submission. Thank you for your consideration and we look forward to hearing your decision on our manuscript.

****

Reviewer #1

MUAC cut offs were largely based on risk of mortality. There are several studies to suggest that MUAC less than 11.5 Cm have higher risk of mortality. However, there is definite need for finding other criteria which has better sensitivity for screening in the community.

Authors’ response: Yes, we believe that this is the objective of our manuscript – to provide information that contributes to the global and local evidence to support the efforts to refine and contextualize criteria for the identification of wasted children, with better sensitivity for community-level screening. 

Reviewer #2

It would be good to provide more rationale for using MUAC in combination with WAZ. WAZ is not an indicator for acute malnutrition. Familiar with health workers, but potential for misclassification, especially if low WAZ is due to low HAZ. Additionally, it loses practical field advantages of WHZ and MUAC in that age in months is required for plotting/calculating WAZ but not needed for MUAC and WHZ. 

Authors’ response: In the discussion section (p17, lines 279-283), we provided context on how WAZ can potentially detect anthropometric deficits including wasting, and near-term mortality. Thus, WAZ can help in improving the diagnostic accuracy of MUAC.

“This is supported by findings from other studies, reporting MUAC with WAZ was able to detect all near-term mortality associated with wasting, concurrent wasting and stunting (WaSt), and other anthropometric deficits (19,29,30). This result is also consistent with studies that have shown the association of WAZ with wasting (weight-for-height), making WAZ a potential anthropometric indicator that could also predict wasting in settings where measurement of height is not possible (31,32).” 

I disagree with the definition provided from line 138 on page 5- that a child should be classified as moderately wasted if their weight-for-age Z-score is -2SD

Authors’ response: The classification that we used to identify moderate wasting using WHZ is if their z-scores fall between -2SD < Z ≤ -3SD which is the definition of the World Health Organization and as adopted in the Philippine Integrated Management of Acute Malnutrition (PIMAM). 

Area under the ROC was used as the indicator of the best-performing MUAC cut-off value- why was something like Youden’s J not used? This would give a better indication of the balance between sensitivity and specificity- the AUROC will only give an indication of the diagnostic accuracy of the test.

Authors’ response: We also conducted an analysis based on the Youden index and we found that the results remain consistent with AUROC as the basis for the best-performing cutoff. Laillou et al. (2014) also used AUROC to determine the optimal cutoff in a similar study . Nonetheless, we revised the full tables in Supplementary Table 1 (S1 Table 1) to add the Youden index, for additional reference.

In table 4, raised MUAC cut-offs are recommended that raise the sensitivity of MUAC to approximately 80%, however this comes at a cost of ~30% lower specificity. Results describe 4.6% severely wasted (WHZ) of 30522= 1419 children; 5.3% (1626 of 30522) severely wasted according to MUAC. Given the numbers presented in the study, sensitivity is 83.23% and specificity is 79.79% for the alternative MUAC, raising the cut-off would result in 1353 correctly classified as true positives, but with a corresponding increase in false positives to approximately 5800. This represents a marked improvement on the number of true positive cases identified by MUAC, but with a correspondingly exponential increase in the number of false positives identified. The much higher absolute number of false positives is due to the relatively low prevalence of the condition in the general population. This point is conceded in the discussion. 

Authors’ response: Thank you for this valuable insight. We agree with this; thus, we integrated this point into the discussion, (p. 15, line 290-293): 

“The results show that with the improvement in the identification of true positives, there is also an exponential increase in the number of false positives, which is due to the low prevalence of wasting in the population.” 

Table 6 presents and interesting analysis of the misclassifications- this on its own would make for an interesting article when assessing current MUAC recommendations. Misclassification often results in eroded trust in health systems, opportunity costs for caregivers and a higher burden of health systems, which needs to be considered in this paper. 

While I agree that nutritionally vulnerable children would benefit from earlier intervention, treatment for severe and moderate acute malnutrition is markedly different from treatment for stunting and related underweight for age. For example, improving dietary diversity as mentioned as a risk factor in these children in the paper may be a good way to treat stunting, and possibly underweight, but will not be affected by programs treating acute malnutrition with F-75 and F-100 based products. Overall it is an interesting idea, however, more consideration needs to be given to the practical implications of the changed MUAC cut-off values presented.

Authors’ response: We agree with this insight that the treatment for acute malnutrition may differ from the treatment for stunting and underweight. We have added nuance to the sentence in the discussion section (p15, line 299-300) to reflect this:

“Hence, these children may benefit from the other appropriate interventions that can be identified upon contact with the health system and which could halt the progression to more severe deterioration of nutrition status and improve the nutrition outcomes of the child."

We emphasize that what is important is that the children have contact with the health system, for them to receive any other appropriate health interventions that they may benefit from. Moreover, in our recommendations, we presented an idea from a study in Cambodia that those children identified using the liberalized cutoff may be assessed further in the health facility prior to enrolment to PIMAM services; to further narrow down the number of false positives. 

Reviewer #3

P4, line 111 - were potential participants able to decline to participate?

Authors’ response: Yes. They can decline to participate. Their participation is voluntary. For clarity, we included in p6, line 163:

“The respondents’ participation is voluntary, and they can refuse to participate.” 

P5, line 124 - was the sample still representative after removal of incomplete cases? Was there anything about this group that may have been different? eg more remote? Does that matter?

Authors’ response: We believe that the sample is still representative after the removal of 3,570 incomplete cases (without anthropometric data). Upon double-checking the data, the distribution across socioeconomic variables and their categories (age, sex, wealth quintile, and type of residence) of the sample with and without the excluded children remains comparable with only <1 percentage point differences. Please see supplementary file 3, and table 1.

P5, line 133 - need refs in this sentence for the method of calculating Sn, Sp, PPV and NPV.

Authors’ response: We added a reference for the calculation of SN, SP, PPV, and NPV (21).

Table 1 - add (months) to age; double check % within subgroups as some do not appear to be correct. If I am misinterpreting I suggest that the table should be reformatted to be clearer.

Authors’ response: The prevalence within subgroup is the calculation of moderate and severe wasting using WHZ and MUAC for each of the subgroup. The sum of the prevalences within each grouping (age, sex, HFAZ, WHZ, wealth quintile, type of residence) should be equal to the overall prevalence of wasting in the topmost row (all sample). The difference of 0.1-2 points in certain groups is due to rounding. 

P9, line 175 - please define the cut-offs for performance for your AUROC. Probably best placed in the methods but if you are going to say they performed poorly what is the criterion for performance.

Authors’ response: We defined as suggested and included the following in the Methods section (p6, line 154-155): 

“Cutoffs with AUROC values of 0.5-0.7 are interpreted as having poor diagnostic performance, 0.7-0.8 as acceptable, 0.8-0.9 as excellent, and 0.9-1.0 as outstanding (23,24).”

Tables 2 and 3 - indication of how Sn and Sp were calculated would help your reader.

Authors’ response: We added a few sentences in the Methods (p5, line 135-138) section to explain the calculation of Sn and Sp. 

“Sensitivity (Specificity) is calculated by dividing the number of individuals with a positive (negative) test result using MUAC by the number of individuals who are identified as wasted (not wasted) using WHZ. Positive (negative) predictive values, on the other hand, are proportion of positive (negative) cases that are truly positive (negative) (21).”

P13, line 234 - I am not clear why this section or Table 6 are included. I cannot see in the methods that this was planned and there appears to be no mention of diet in the methods at all. Similarly the discussion of this content. If it is to be included there needs to be better alignment through the methods and ideally the same content provided for all children for more transparent comparison.

Authors’ response: We understand that the said section or table may not be fit in the overall research question. Thus, we moved the full table in the supplementary file as this was only assessed to guide the discussion of the results. Furthermore, we declared in the Methods section (p6, lines 144-148) the details of this assessment: 

“Furthermore, we also assessed the characteristics (age, sex, underweight status, stunting status, dietary intake, diversity score, minimum meal frequency, and minimum acceptable diet) of children incorrectly classified as wasted vis-à-vis WHZ to provide additional information in the discussion (See Supplementary File 2 (S2)). Full details on the data collection methodology employed for the dietary component can be found in the survey report (20).”

---

## [Decision Letter · Decision Letter 1]

22 Nov 2024

Determining an optimal case definition using mid-upper arm circumference with or without weight for age to identify childhood wasting in the Philippines

PONE-D-24-23038R1

Dear Dr. Casas,

We’re pleased to inform you that your manuscript has been judged scientifically suitable for publication and will be formally accepted for publication once it meets all outstanding technical requirements.

Kind regards,

Guy Franck Biaou ALE, PhD

Academic Editor

PLOS ONE

Additional Editor Comments (optional):

Reviewers' comments:

Reviewer's Responses to Questions

**Comments to the Author**

1. If the authors have adequately addressed your comments raised in a previous round of review and you feel that this manuscript is now acceptable for publication, you may indicate that here to bypass the “Comments to the Author” section, enter your conflict of interest statement in the “Confidential to Editor” section, and submit your "Accept" recommendation.

Reviewer #1: All comments have been addressed

Reviewer #3: (No Response)

2. Is the manuscript technically sound, and do the data support the conclusions?

Reviewer #1: Yes

Reviewer #3: Yes

3. Has the statistical analysis been performed appropriately and rigorously? 

Reviewer #1: Yes

Reviewer #3: Yes

4. Have the authors made all data underlying the findings in their manuscript fully available?

Reviewer #1: Yes

Reviewer #3: No

5. Is the manuscript presented in an intelligible fashion and written in standard English?

Reviewer #1: Yes

Reviewer #3: Yes

6. Review Comments to the Author

Reviewer #1: Although higher cutoffs is definitely going to improve sensitivity and will be useful still WFH will be helpful in identifying children having multiple anthro deficit

Reviewer #3: Thank you for the time taken to address the comments provided at review. Please add a footnote to table 1 to account for differences due to rounding.

7. PLOS authors have the option to publish the peer review history of their article (what does this mean?). If published, this will include your full peer review and any attached files.

Reviewer #1: **Yes: **Dr Praveen Kumar

Reviewer #3: No

---

## [Editor Report · Acceptance letter]

12 Dec 2024

PONE-D-24-23038R1 

PLOS ONE

Dear Dr. Casas, 

I'm pleased to inform you that your manuscript has been deemed suitable for publication in PLOS ONE. Congratulations! Your manuscript is now being handed over to our production team.

Kind regards, 

on behalf of

Dr. Guy Franck Biaou ALE 

Academic Editor

PLOS ONE